# Roll-to-roll, high-resolution 3D printing of shape-specific particles

Jason M. Kronenfeld[1], Lukas Rother[2], Max A. Saccone[3], Maria T. Dulay[2] & Joseph M. DeSimone[3✉]

Particle fabrication has attracted recent attention owing to its diverse applications in bioengineering[1,2], drug and vaccine delivery[3–5], microfluidics[6,7], granular systems[8,9], self-assembly[5,10,11], microelectronics[12,13] and abrasives[14]. Herein we introduce a scalable, high-resolution, 3D printing technique for the fabrication of shape-specific particles based on roll-to-roll continuous liquid interface production (r2rCLIP). We demonstrate r2rCLIP using single-digit, micron-resolution optics in combination with a continuous roll of film (in lieu of a static platform), enabling the rapidly permutable fabrication and harvesting of shape-specific particles from a variety of materials and with complex geometries, including geometries not possible to achieve with advanced mould-based techniques. We demonstrate r2rCLIP production of mouldable and non-mouldable shapes with voxel sizes as small as $2.0 \times 2.0 \ \mu m^2$ in the print plane and $1.1 \pm 0.3 \ \mu m$ unsupported thickness, at speeds of up to 1,000,000 particles per day. Such microscopic particles with permutable, intricate designs enable direct integration within biomedical, analytical and advanced materials applications.

Particles on the scale of hundreds of micrometres to nanometres are ubiquitous key components in many advanced applications including biomedical devices[1,2], drug-delivery systems[3–5,15], microelectronics[12] and energy storage systems[16,17], and exhibit inherent material applicability in microfluidics[6,7], granular systems[8,9] and abrasives[14]. Approaches to particle fabrication inherently have trade-offs among speed, scalability, geometric control, uniformity and material properties.

Traditional particle fabrication methods range from milling and emulsification techniques to advanced moulding and flow lithography, and approaches can be classified as either bottom-up or top-down. Bottom-up particle fabrication approaches, best exemplified by grinding and milling[18], emulsification[19], precipitation[20], nucleation-and-growth[21] and self-assembly[5,10,11] techniques, can have high throughput but lead to heterogeneous populations of granular particles with limited control over shape and uniformity. To address the geometric shortcomings of bottom-up approaches, top-down particle fabrication methods such as direct lithography[10,22], single-step roll-to-roll soft lithography[23,24] and multistep moulding[4] have been employed.

Scalable particle moulding approaches, such as particle replication in non-wetting templates (PRINT) and stamped assembly of polymer layers (SEAL), incorporate lithographic approaches to attain two-dimensional (2D) geometric control[4,24]. PRINT utilizes a non-wetting fluoropolymer layer to facilitate rapid fabrication of isolated micro- and nanoparticles with demonstrable precise control over shape, size, surface functionalization and fillers such as drugs, proteins or DNA/RNA[24,25]. Detailed in vitro studies of these particles have elucidated shape-dependent tendencies of cellular uptake and enhanced localized cargo release[24–26]. Moreover, in vivo studies have shown the significant role played by particle size, shape, charge, surface chemistry and particle deformability on biodistribution via multiple different dosage forms (injection and inhalation)[27–29]. Extending the PRINT technology, the stacking of moulded particles enables more complex particle geometries as exemplified by SEAL[4]. Harvested moulded sections are welded together to gain three-dimensional (3D) fabrication control, yielding demonstratable pulsatile-release, drug-delivery vehicles. The trajectory and demonstrated application potential of these technologies lays the groundwork for future methods of fabricating advanced particles.

For example, continuous-flow lithography (or optofluidic fabrication) produces particles as a photopolymerizable resin flows through a fluidic channel, curing in 2D to 3D geometries[30,31]. The stop-polymerize-flow technique has been demonstrated to achieve quasi-continuous fabrication of 2D to 2.5D geometries (anisotropic properties on a 2D-defined shape)[32]. Deterministic deformation based on microfluidic flow can further enable the fabrication of concave-surface geometries, previously demonstrated at the rate of 86,400 particles per day[31]. Furthermore, additional dimensional control processes may be introduced to create Janus particles (particles whose surfaces have two or more distinct physical properties), nanoporous meshes using sacrificial additives or porogens or micropatterning via secondary chemical coating or formation control steps[2,33,34].

One remaining major engineering challenge is to develop a particle fabrication technique that simultaneously enables all dimensions of micron-scale 3D geometric control, complexity, speed, material selection and permutability. Herein we introduce a scalable, high-resolution 3D printing technique for particle fabrication based on a roll-to-roll form of continuous liquid interface production (r2rCLIP). We demonstrate r2rCLIP using single-digit, micron-resolution optics

[1]Department of Chemistry, Stanford University, Stanford, CA, USA. [2]Department of Radiology, Stanford University, Stanford, CA, USA. [3]Department of Chemical Engineering, Department of Radiology, Stanford University, Stanford, CA, USA. [✉]e-mail: jmdesimone@stanford.edu

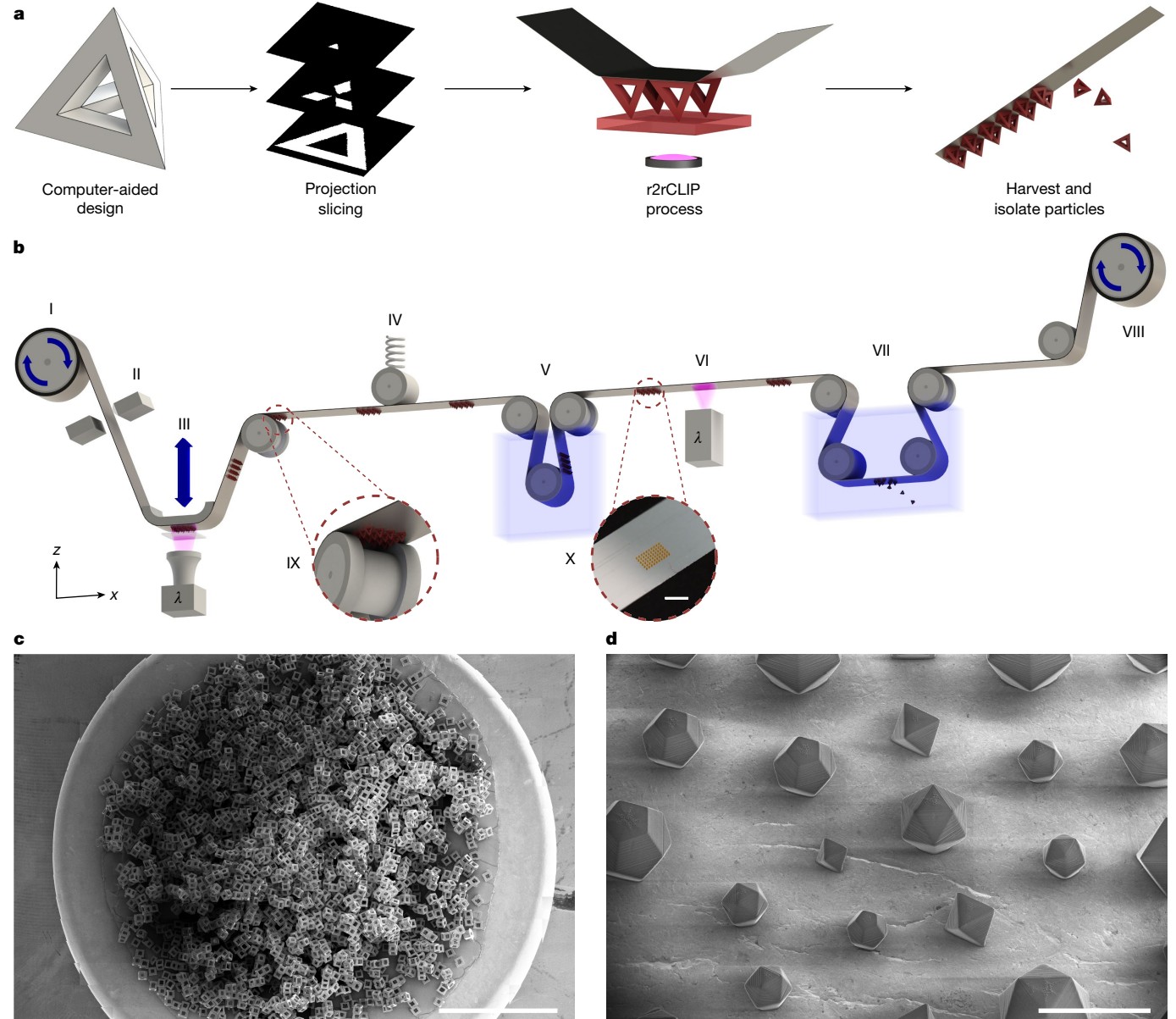

**Fig. 1 | r2rCLIP is a rapid fabrication process for particles with complex geometries. a**, r2rCLIP is a quasi-continuous technique wherein a 3D geometry of simple to complex nature is designed and subsequently sliced into 2D images. These images are then used to fabricate 3D geometries from a photopolymerizable resin in a roll-to-roll process. **b**, Diagram of experimental r2rCLIP setup wherein an aluminium-coated PET film is unrolled from a feed roll (I) and mechanically braked (II) to provide tension before passing over a high-precision *z* stage and CLIP assembly (III). A designed geometry is projected through a Teflon AF window into a vat of photopolymerizable resin. The geometry materializes onto the film and the stage pulls in the *z* direction to direct vertical part formation. Once materialized, the particles on film are passed under a spring-tensioning system to maintain relative substrate positioning during stage movement (IV). The film is then passed through a cleaning step (V) before secondary curing (VI) and immersion in a non-ionic surfactant solution within a heated sonication bath and a razor blade to induce delamination (VII). The film is finally collected on a second roller with a stepper motor that provides translational movement throughout the process (VIII; Extended Data Fig. 1). Insets show a graphic of particle clearance over a guide roller (IX) and an image of particles on the film post cleaning (X). **c**, This scalable process is demonstrated by the production of around 30,000 hollow cube particles observed in a set of computer-stitched scanning electron microscopy (SEM) images. **d**, Octahedrons, icosahedrons and dodecahedrons with unit cell size ranging from 200 to 400 µm printed within a singular printed array. **c**,**d**, Samples printed from the HDDA–HDDMA system and coated with Au/Pd (60:40) before SEM imaging. Scale bars, 3 mm (**b**,**c**), 500 µm (**d**).

in combination with a continuous roll of film in lieu of a static platform, enabling fast, rapidly permutable fabrication and harvesting of particles with a variety of materials and complex geometries (Fig. 1).

Continuous liquid interface production is an additive manufacturing technique that uses digital light processing (DLP) to project videos of 2D images describing 3D models into a vat of photopolymerizable resin. The resolution of this technique has improved from 50 to 4.5 µm, as well

as providing speeds of up to 3,000 mm h$^{-1}$ (refs. 35–38). CLIP utilizes a 385 nm ultraviolet light-emitting diode (LED) and digital micromirror device to simultaneously pattern an array of actinic photons, activating photo-initiators dissolved in liquid resin and inducing radical polymerization in each printed voxel. The CLIP technique is distinguished by the introduction of an oxygen-induced, photopolymerization-inhibited 'dead zone' between the photocurable resin and an optically clear

**Table 1 | Experimental curing parameters for high-resolution resins utilized in particle fabrication**

| Resin | Components[a] | Minimum resolved bridge thickness (µm) | $D_p$ (µm) | $E_{crit}$ (mJ cm⁻²)[b] |
|---|---|---|---|---|
| HDDA–HDDMA[39] | HDDA with 0.5 wt% HDDMA, 5.0 wt% PPO, 0.5 wt% Sudan I | 1.1±0.3 | 8.0±0.4 | 5.2±0.3 |
| PEGDA₄₀₀ (ref. 42) | PEGDA₄₀₀ with 1.0 wt% PPO, 0.5 wt% Sudan I | 3.4±0.5 | 12.1±0.5 | 8.5±0.3 |
| PEGDMA₅₅₀ | PEGDMA₅₅₀ with 2.5 wt% TPO, 0.4 wt% BLS1326 | 8.1±0.3 | 6.3±0.3 | 2.0±0.3 |
| TMPTA[36] | TMPTA with 2.5 wt% TPO, 0.4 wt% BLS1326 | 4±1 | 63.0±0.9 | 2.93±0.05 |
| HDDA–ceramic mix | SIL 30 component A with 47.0 wt% HDDA, 2.5 wt% PPO, 0.5 wt% Sudan I, 0.3 wt% HDDMA | 4.3±0.5 | 10.2±0.7 | 14±1 |
| PR48–Clear | 40 wt% Allnex Ebecryl 8210, 40 wt% Sartomer SR 494, 0.4 wt% Esstech TPO+, 20 wt% Rahn Genomer 1122, 0.2 wt% Mayzo OB+ | 4±1 | 51±2 | 12.8±0.5 |
| KeySplint Hard | 10–25 wt% 2-phenoxyethyl methacrylate, 10–25 wt% isobornyl methacrylate, 3 wt% 2-hydroxyethyl methacrylate, 3 wt% TPO | 6.8±0.3 | 89±8 | 4.9±0.3 |

[a]Commercial resins reported as disclosed components in respective manufacturers' safety data sheets.

[b]Based on LED intensity reported by the In-Vision Helios DLP Engine that, as such, accounts for light intensity incident only at the Teflon AF window and not necessarily at the resin–dead zone interface.

As previously reported[40,41], model-predicted parameters ignore fragility and ability to resolve small features and should be used explicitly as an informed baseline parameter when considering dynamic printing parameters within each voxel (greyscaling to reduce overcure and accumulated dosage, optical proximity correction and so on). Error reported as plus or minus standard error of parameters; $n$ = 101, 32, 38, 33, 51, 49 and 15 bridges analysed for each resin composition, respectively (top to bottom in table), with multiple measurements taken per bridge to minimize error of measurement. PPO, phenylbis(2,4,6-trimethylbenzoyl) phosphine oxide; TMPTA, trimethylolpropane triacrylate; PEGDA, poly(ethylene glycol) diacrylate; PEGDMA, poly(ethylene glycol) dimethacrylate.

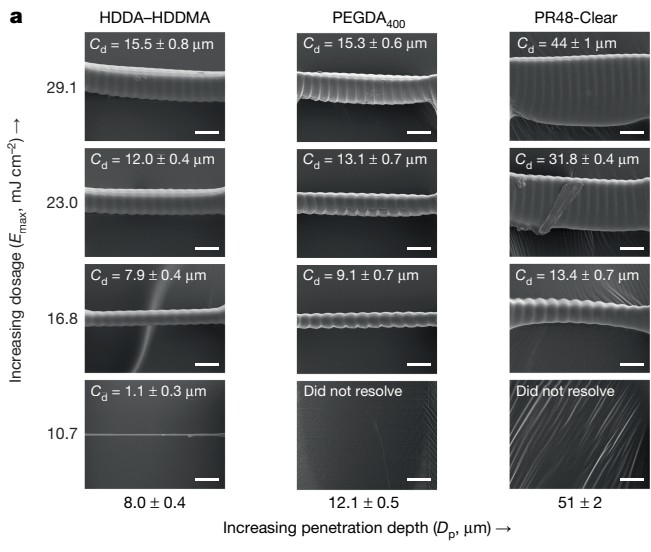

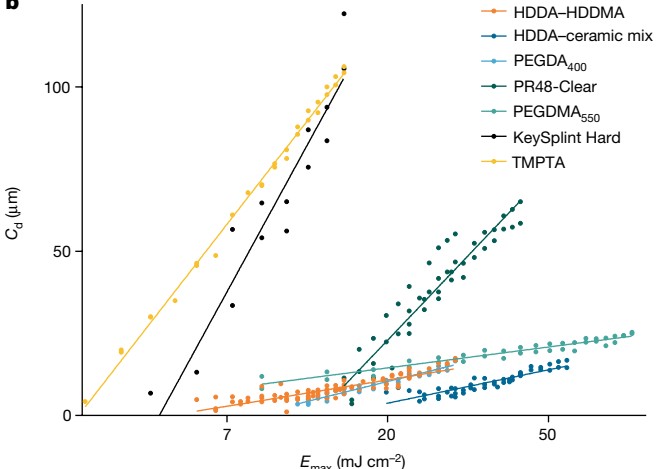

**Fig. 2 | r2rCLIP is amenable to a range of high-resolution in-house and commercial materials with high-precision optimization. a**, The bridging method enables working curve determination of resin-curing properties, as demonstrated for several bridge series from resins of increasing penetration depth at constant dosage and corresponding measured cure depth. Ridging artefacts coincide with pixel pitch at 6 µm spacing. Exposure measurement bridges coated with Au/Pd (60:40) before SEM imaging. **b**, Determination of intrinsic penetration depth and critical cure dosage. A lower slope correlates with greater analytical cure depth control at a given dosage ($E_{max}$), as well as with a lower propensity for fluctuations in exposure to result in major changes in cure depth ($C_d$). Scale bars, 15 µm.

vat window (Teflon amorphous fluoropolymer (AF) 1600 or 2400), effectively obviating any delamination step (Extended Data Fig. 2. and Supplementary Note 1). Lack of adherence, or glueing, of the growing particle onto the window facilitates fabrication of fragile green parts, such as thin struts on hollowed particle geometries, while maintaining high throughput speeds[35,36]. This technique is demonstrably versatile for a broad range of polymer chemistries, functionalization, fillers and multimaterial platforms[35,38]. High-resolution CLIP is used herein to obtain geometric control for the scalable fabrication of particles in the sub-200-µm regime with resin-dependent, layer-wise control down to single-digit-micron range and 2.00 × 2.00 µm² $xy$ resolution.

To achieve a rapid and fully automated particle-printing process we substituted the conventional static build plate of a high-resolution CLIP printer with a continuous-film, modular, roll-to-roll system. This enables semicontinuous printing and automated in-line postprocessing including cleaning, postcuring and harvesting (particle liftoff). An aluminium-coated polyethylene terephthalate (PET) film was chosen as the primary film substrate to maintain particle adhesion during printing at a level above in situ orthogonal resin reflow forces and normal suction forces, still allowing for delamination from film without fracture during harvesting (for additional substrates tested see Supplementary Note 2).

Complementary to film integration for particle printing, we constructed a high-resolution CLIP setup to fabricate fine particle features that achieves single-digit-micron optical resolution (2.00 × 2.00 or 6.00 × 6.00 µm² depending on desired build area) in the $xy$ plane. Voxel definition further depends on vertical resolution, dependent on stage movement repeatability (±0.12 µm), depth of focus of the optical setup (for example, 30 µm for 2.00 × 2.00 µm² setup) and

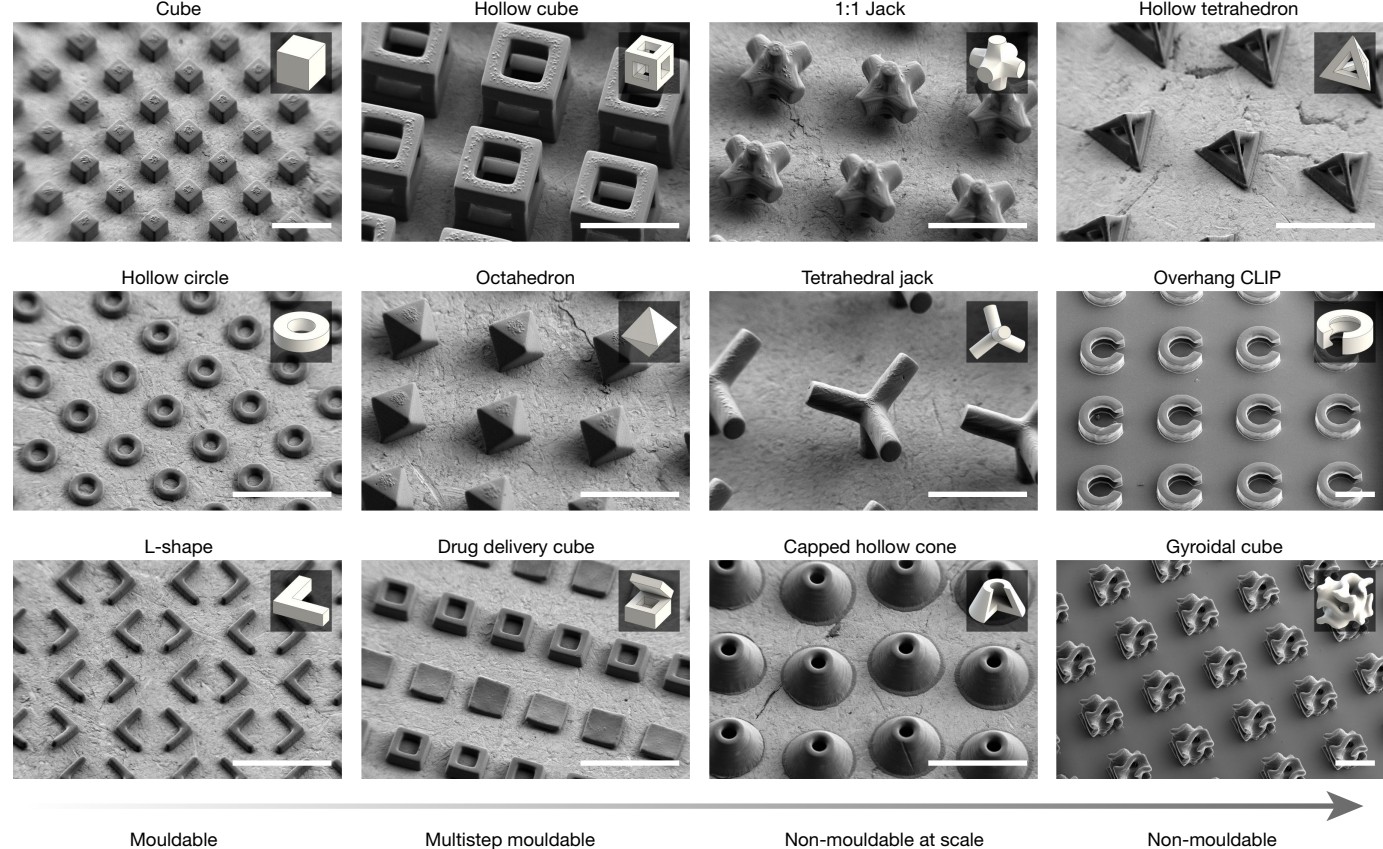

| Cube | Hollow cube | 1:1 Jack | Hollow tetrahedron |
| Hollow circle | Octahedron | Tetrahedral jack | Overhang CLIP |
| L-shape | Drug delivery cube | Capped hollow cone | Gyroidal cube |

| Mouldable | Multistep mouldable | Non-mouldable at scale | Non-mouldable |

**Fig. 3 | SEM images of mouldable to non-mouldable geometries fabricated by r2rCLIP.** Particles were fabricated using the HDDA–HDDMA system and informed exposure intensities obtained from bridge fitting data (Fig. 2 and Table 1), washed as described and coated with a 60:40 Au/Pd before SEM observation. Insets show a rendering of each respective geometry for reference. Capped hollow cone inset shown as quarter cut-through for clarity. Scale bars, 250 μm.

resin physical properties (refraction and diffraction of light, penetration depth and critical exposure dose for gelation; Table 1, Fig. 2 and Supplementary Note 3).

Previous work has studied surface and resolution optimization in photopolymerization-based 3D printing systems[39]; achieving $z$ resolution below 25 μm remains a challenge due to intrinsic resin penetration depth and overcuring from accumulated dosages[40–42]. To fabricate optimal, complex particle geometries a resin system must be designed to achieve high $z$ resolution; a 1,6-hexanediol diacrylate–1,6-hexanediol dimethacrylate (HDDA–HDDMA)-based system was previously described as achieving up to 4 μm vertical resolution[39]. We utilize this resin system herein and adopt an analytical bridging technique to measure intrinsic resin properties, as opposed to the common glass slide method[40,42,43] which does not analytically describe in situ high-resolution CLIP as accurately. Our HDDA–HDDMA resin has a characteristic penetration depth of 8.0 ± 0.4 μm and experimentally resolved a minimum unsupported bridge thickness of 1.1 ± 0.3 μm. We characterized several additional high-resolution custom and commercial resin compositions, which are also compatible with r2rCLIP and may be substituted depending on materials requirements, desired vertical resolution and application (Table 1 and Fig. 2). Notably, unsupported film bridges characterized in the curing assay are thin (under 100 μm, relevant to particle fabrication) and resolve proximal to the dead zone, introducing periodic artefacts ascribed to fluctuations in light intensity between pixels. Surface irregularities may further be attributed to either resin reflow (elongated lines) or cavitation (bubbles) and may be addressed with optimization. Resin parameterization and optimization are essential in regard to vertical resolution determination for fabrication limitations; resins with greater characteristic penetration depth are not as amenable to thin vertical geometric features.

To demonstrate the potential of r2rCLIP in the fabrication of dimensionally complex structures we designed a range of shapes with increasing geometric complexity using computer-aided design. These designs not only mirror those created by previous 2D fabrication and multistep moulding techniques[4,24] but also include several geometries that cannot be moulded, exemplifying the unique capabilities of our approach (Fig. 3). Herein we categorize geometric complexity on a spectrum ranging from shapes that can be moulded at scale to those that cannot. Mouldable geometries are defined to be plausibly fabricated at scale in a single step using a uniaxial die draw, core and cavity. Geometries increase in moulding complexity (and subsequently decrease in mouldability at scale) if a theoretical moulding approach requires an increasing number of parting lines, ejector pins and angles and extensive alignment or contains non-mouldable negative internal spaces. In addition, thin or sharp geometric features may introduce moulding complications and part anisotropy due to, for example, flash, short shot, shrinkage or air pockets exacerbated at the micron scale (Supplementary Note 4)[44]. It should be noted that it is plausible to couple a multistep moulding process with a sacrificial etching step to achieve some geometries deemed non-mouldable in this work, although without a high degree of reproducibility given mould alignment requirements.

One significant benefit of using the r2rCLIP method for particle fabrication is its inherent mouldless process, which enables changing of fabricated geometries within or between arrays based solely on optimized printing parameters. This means that a wide variety of particle geometries can be produced without needing to alter the setup, as would be necessary with previous particle fabrication methods

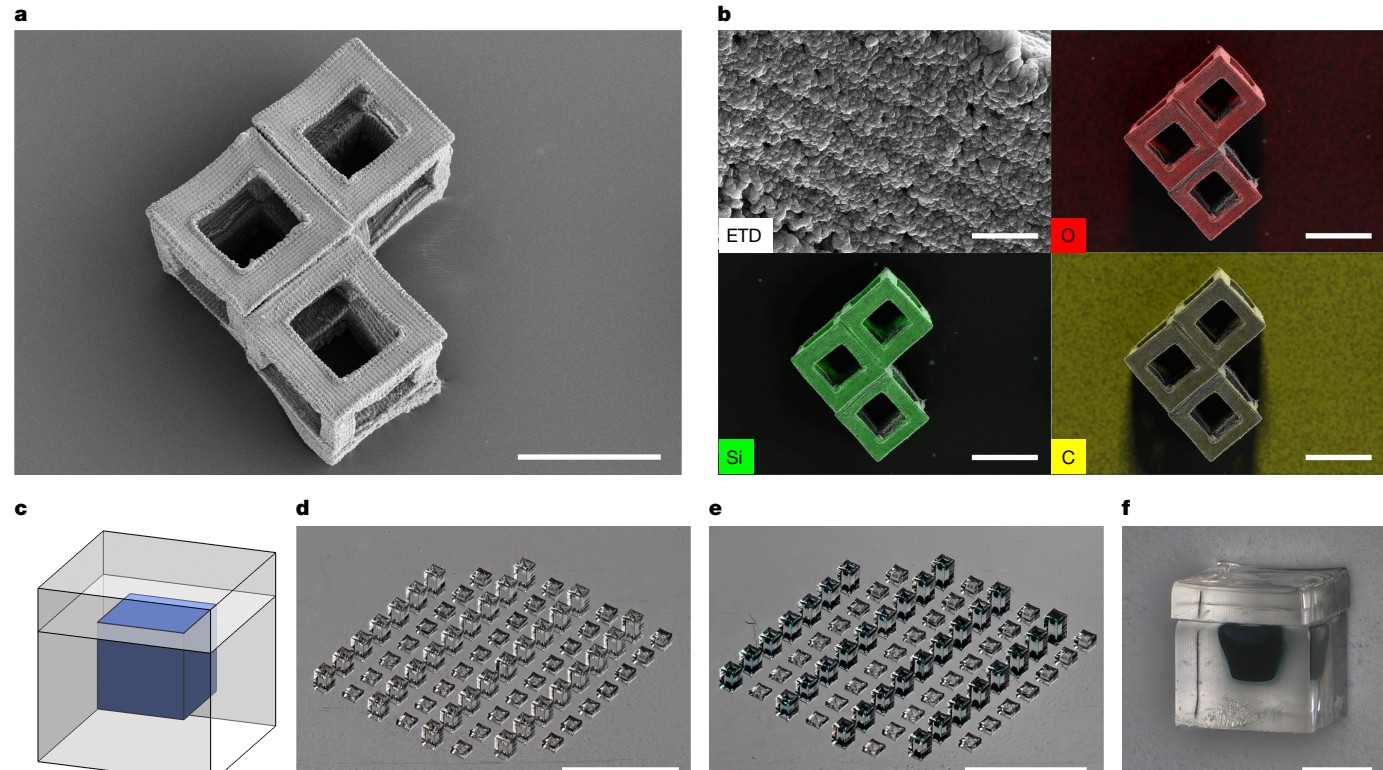

**Fig. 4 | Particles fabricated via r2rCLIP enable a range of applications including ceramic particles and drug delivery. a**, Hollow ceramic cubes formed from pyrolysis of HDDA–ceramic mix resin. **b**, EDS analysis of the surface of a hollow ceramic cube (top left) showing uniform distribution of silicon and oxygen, quantified as $30 \pm 1\%$ silicon, $35 \pm 1\%$ oxygen and $35 \pm 2\%$ carbon by normalized mass. Elemental distribution of O, Si and C (top right, bottom left and bottom right, respectively) overlaid on secondary electron image of the hollow cubes. **c**,**d**, Drug-delivery cubes may be designed to meet the goals of payload volume, release profile, material and so on (**c**) and fabricated via r2rCLIP (**d**) ($PEGDMA_{550}$ material, for example). **e**,**f**, Devices may be then filled, as demonstrated with trypan blue dye for visualization (**e**), and subsequently capped (**f**). Scale bars, 100 μm (**a**), 5 μm (**b**, top left), 100 μm (**b**, other three images), 3 mm (**d**,**e**), 200 μm (**f**).

(for example, mould interchange). This flexibility is particularly beneficial when needing to adjust geometric requirements, such as when fabricating precise ratios of heterogeneous mixtures of polydisperse particles (Fig. 1d).

To demonstrate the scalability afforded by r2rCLIP we fabricated approximately 30,000 hollow cube-shaped particles of 200 μm width and high reproducibility (Fig. 1c; $96 \pm 1\%$ fabrication success rate, $n = 300$; $-10 \pm 20\%$ average relative error from nominal strut feature size, $n = 300$). Whereas optimized particle array (up to 16.4 mm² for 2 μm or 147.5 mm² for 6 μm resolution) fabrication speed is subminute, gram-scale production (thousands to millions of particles) necessitates the removal of time-consuming, manual manipulation steps. Previously the slow step of particle production involved the manual replacement of build substrate (requiring $4 \pm 2$ min for manual manipulation between high-resolution CLIP print jobs, $n = 6,436$; Supplementary Note 5). Replacing this manual manipulation step with mechanical substrate translation shifts the rate-limiting step to particle fabrication time—an inherent advantage of the r2rCLIP technique. For instance, fabrication of 1 million 200-μm-unit octahedrons (equal to approximately 1.4 g) would require just over 1 day with demonstrated array fabrication speeds of up to 38 s print duration with 26 s interprint delay (Supplementary Note 6). The r2rCLIP platform thus enables a new design application of particle fabrication in a wide range of accessible geometries, materials and batch sizes. r2rCLIP is a modular process that can thus be adapted to include additional steps in series such as coating, filling or sterilization, as well as additional postharvesting treatments such as devolatilization, electroless deposition or functionalization. The high throughput of r2rCLIP has direct implications

for industrial-scale production of microdevices such as microrobots and cargo delivery systems.

As an example, this system is amenable to the production of ceramic materials. Preceramic resins can be used to mass produce technical ceramic particles, with potential applications in chemical mechanical planarization techniques as slurry components, conductive particles, in microtools, microelectromechanical systems or waveguides, enabling industrial applications such as electronics, telecommunications and healthcare[13]. As an example, we created 200 μm particles from a HDDA–preceramic mix and pyrolysed them in nitrogen at 800 °C to produce 103 μm hollow ceramic particles of feature size 25 μm (Fig. 4a). Energy-dispersive X-ray spectroscopy (EDS) analysis of these particles showed uniform composition distribution of O, Si and C (Fig. 4b). With subsequent annealing up to 1,400 °C in nitrogen, phases including $Si_3N_4$ and $SiO_2$ can be achieved depending on the precursor material and processing conditions (Extended Data Fig. 3 and Supplementary Note 7). Future research can investigate the effectiveness of this process with different preceramic formulations and explore their potential applications.

One further application enabled by r2rCLIP is the creation of hydrogel particles, which can be used as drug-delivery vessels. These particles can be filled to achieve adjustable, gradient or pulsatile-release profiles in a singular injection, as previously demonstrated for the SEAL process[4,45,46]. Previous studies have explored the development of suitable photopolymer resin systems and the impact of materials biocompatibility, cytotoxicity, shape and size on localization and delivery, enabling the creation of bioscaffolds and delivery manifolds[5,15,23,25,28,45–49]. This opens new possibilities for the fabrication of hydrogel particles for

drug delivery but lacks a permutable, scalable fabrication process. As a proof of concept we have fabricated hydrogel cubes of 400 μm unit size, manually filled with around 8 nl of representative cargo postprinting and subsequently topped with a hydrogel cap (Fig. 4c). Future research can build on previous studies on drug-delivery vehicle kinetics, leveraging the adjustable properties of molecular weight and wall thickness to achieve a programmable pallet of cargo release.

Furthermore, amine-functionalized polymer end groups could be added to facilitate postfunctionalization with fluorophores, enabling the potential to integrate single-particle, one-pot analytical techniques to localize signal for better detection. Smaller unit scale geometries and additional materials such as metals may even be achieved through thermal conversion postprocessing that could lead to roughly 70% reduction in feature size[50], which would bring our current $xy$ resolution onto the nanometre scale. Future system improvement work can explore print and speed optimization, soluble film coatings, cleaning and particle-harvesting methods.

The mechanical and material versatility, ranging from hard ceramics to soft hydrogels, could support the creation of Janus particle properties and smart materials and aid in fundamental studies in materials and granular physics. Although the system requires a photopolymerizable component, it can accommodate weak, green-state particles enabling mixed, dual-curing systems containing a non-photopolymerizable component addressed in postprocessing. This flexibility allows for tunable particle materials properties dependent on the resin system, enabling a variety of particles with different mechanical properties to meet application requirements.

Herein we present a new, roll-to-roll, high-resolution, continuous liquid interface production technique capable of mass production of particles up to 200 μm at up to 2.0 μm feature resolution. Optical design of both printer and resin optimization enables printing of objects with up to single-digit-micron unsupported $z$ resolution. Rapid permutability, complex 3D fabrication capabilities and inherent amenability to a wide variety of resin chemistries are demonstrated in the fabrication of mouldable, multistep mouldable and non-mouldable particle geometries. Moreover, rapid particle production enables gram-scale potential yield within a period of around 24 h for sub-200-μm units. This scalable particle production technique has demonstrated fabrication potential over a wide range, from ceramic to hydrogel manifolds, with subsequent potential application in microtools, electronics and drug delivery.

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

## Methods

### Materials

All compounds were purchased from Sigma-Aldrich, unless otherwise noted, and used without further purification. An HDDA–HDDMA resin formulation was created based on previous literature results for minimization of layer thickness while maintaining surface homogeneity[39]. The formulation consists of 0.5 wt% HDDMA, 5.0 wt% PPO as photo-initiator and 0.5 wt% Sudan I as ultraviolet absorber. The formulation was filtered through a 0.22 µm nylon filter (no. 1470347, Sterlitech Corporation) immediately before use. The TMPTA formulation consisted of TMPTA with 2.5 wt% TPO and 0.3 wt% BLS1326. The PEGDA$_{400}$ formulation consisted of PEGDA (Mn, 400; no. 01871-250, PolySciences, Inc.), 1.0 wt% PPO and 0.5 wt% Sudan I. The preceramic resin formulation consisted of 49.8 wt% SIL 30 Component A (Carbon, Inc.), 47.0 wt% HDDA, 2.5 wt% PPO, 0.5 wt% Sudan I and 0.25 wt% HDDMA. The demonstration drug-delivery cubes were fabricated from PEGDMA (Mn, 550) with 2.5 wt% TPO and 0.4 wt% BLS1326 and postprint filled with 10 v/v% methylene blue in glycerol (no. G33-500, Fisher Chemical). Commercial resins tested included PR48 (PR48-Clear Resin, CPS Polymers) and KeySplint Hard (no. 2421146, Henry Schein).

### High-resolution CLIP setup

The high-resolution CLIP printer design has been described previously[36]. The system was modified to include a high-precision $z$ stage (no. KVS30/M, ThorLabs) and a 385 nm DLP light engine with either a 2.00 µm lens (9000 Firebird Light Engine, In-Vision Technologies AG) or a 6.00 µm lens (9000 Helios Light Engine, In-Vision Technologies AG). An absorptive neutral density filter (no. NE210B, ThorLabs) was placed in the optical path of the Helios engine to reduce relative intensity to a controlled, analytical range. The printer was controlled by custom software written in C++. Computer-aided design was performed in SolidWorks 2022, and slicing in Autodesk Netfabb Premium 2024.

### Mechanical setup

A 12.7-mm-wide, aluminium-coated polyethylene terephthalate film (no. 48-2F-1M, CS Hyde) acted as the primary printing substrate. The film was tensioned throughout the printing process via a Teflon-covered adjustable brake (15-1F-.5, CS Hyde) and a spring-loaded tensioner to ensure a flat surface over the print head. Particles printed directly onto the film were guided over 3D-printed rollers, which touch only the outside of the film in sections where no particles are printed. The film passed from a feedstock, over the print head, through a cleaning bath and a secondary LED cure step when applicable (no. LZ1-10UBN0-00U4, ams-OSRAM), through a heating (50 °C for HDDA–HDDMA print runs) and sonication bath with a stainless steel razor blade for harvesting of particles and was finally loaded onto a collection roller. The collection roller was coupled with a geared stepper motor (no. 17HS19-1684S-PG100, OSM Technology Co.) to guide film advancement, controlled by a motion control board (Octopus v.1.1, Bigtreetech) in serial communication with the printer interface.

### Particle harvesting

Particles were printed directly onto a film substrate, immersed for 1 min in isopropanol and harvested in-line via sonication and heating (DK-200H, DK Sonic) in 2 wt% Pluronic F-127 in water with a stainless steel razor blade to ensure delamination. Exact processing could be tuned to individual resin systems (Supplementary Note 6). Particles were collected by gravity filtration over a 30-µm-hole stainless steel mesh (400 Mesh, Uxcell).

### Polymer-derived ceramics processing

The HDDA–ceramic mix resin was prepared by mixing individual components and subsequent centrifuging at 600 rpm for 10 min. The upper phase, about 75% of the total volume, was collected and centrifuged again. This was repeated three times to remove larger particulates.

HDDA–ceramic mix particles were pyrolysed in an alumina boat within a quartz tube furnace (Thermo Scientific Lindberg Blue M) under nitrogen gas flow. A ramp rate of 5 °C min$^{-1}$ was used, with 2 h isothermal holds at 400 and 800 °C. After purging the furnace three times with house nitrogen, house nitrogen was flowed at 150 standard cubic centimetres per minute (sccm) and atmospheric pressure for the duration of the pyrolysis treatment.

To demonstrate the fabrication of technical ceramics, selected samples of pure SIL 30 resin (containing the Si-containing component of the HDDA–ceramic mix resin) were annealed at higher temperatures for X-ray diffraction (XRD) analysis; these samples were first annealed at 800 °C, as described above, then subsequently annealed in nitrogen for 2 h at 1,200 °C followed by 2 h at 1,400 °C, using a ramp rate of 5 °C min$^{-1}$ and an alumina tube. Between each annealing step, XRD patterns were collected for samples that had been crushed to powder with a mortar and pestle. In addition, samples of the HDDA–ceramic mix resin were pyrolysed at 1,400 °C in nitrogen for a total of 8 h, using a ramp rate of 5 °C min$^{-1}$ and a house nitrogen flow rate of 150 sccm. Samples were first annealed at 1,400 °C in nitrogen for 2 h, following which they developed a dark grey colour. Samples were then crushed with a mortar and pestle and annealed for a further 6 h at 1,400 °C under the same conditions, after which they developed a light grey colour. XRD patterns were collected for the resulting powder samples.

### Cargo delivery cube fabrication

To produce cargo delivery cubes, $400 \times 400 \times 300$ µm$^3$ boxes were fabricated from the PEGDMA$_{550}$ formulation to reflect previous drug-delivery research[4]. These boxes contained a negative inner space of $200 \times 200$ µm$^2$ at a wall thickness of 100 µm. After briefly rinsing out of uncured resin with isopropanol, the box bases were placed under a stereoscope (SM-1TSZZ-144S-16M3, AmScope) and filled manually with 10 v/v% methylene blue in glycerol via a fused-silica capillary (no. 1068150015, Polymicro Technologies), threaded and glued into a syringe needle (no. 329461, BD). The box lid was manipulated manually to sit on top of the lower section, briefly coated with uncured PEGDMA$_{550}$ resin and cured to seal. Future studies could utilize micromanipulators, cameras and/or a picolitre liquid dispenser to fill and align/cap arrays of boxes separately to eliminate the imprecision of manual manipulation. Furthermore, wall thickness, unit size, geometry and oligomer molecular weight can be iteratively tuned to potentially modulate release kinetics and localization.

### Characterization

Resin cure tests were performed by the bridging method on the r2rCLIP printer: a series of bridges for each resin of interest were fabricated in situ (90 µm length; Extended Data Fig. 2). HDDA–HDDMA bridge datasets were expanded to lengths of 90 and 180 µm to increase robustness over the particle size regime. The resulting $z$-bridge thickness was measured by SEM and recorded as cure depth, $C_d$. PEGDMA$_{550}$ bridges were analysed via optical imaging to circumvent the impact of vacuum-induced shrivelling on fitting data. Resin intrinsic penetration depth, $D_p$, and critical cure dosage, $E_{crit}$, were interpolated by fitting a working curve to cure depth as a function of dosage for an analytical range appropriate for particle layer fabrication, as previously described[40].

Optical imaging was performed on an Olympus DSX1000 (Evident Scientific) with corresponding analysis software. SEM was performed on a Thermo Fisher Scientific Apreo S LoVac scanning electron microscope (1–10 kV, 50–400 pA, nominal working distance 10 mm, using an Everhart–Thornley detector) with a retractable XFlash 6 | 60 SDD EDS detector with Esprit analysis software. SEM samples were investigated with or without 60:40 Au/Pd coating depending on sample size, substrate and required analysis—indicated as such in the respective figures. Images were analysed on ImageJ 1.54 and visualized in InkScape 1.3.

Statistical analysis and visualization were performed using GraphPad Prism 10.1.0, with parameters fit using least-squares regression without weighting. Error is presented as either standard error of parameters assuming an asymptotic confidence interval for model fit parameter output or standard deviation for sampling sets.

X-ray diffraction experiments on powder samples were performed on a PANalytical Empyrean diffractometer using an iCore PreFIX module and GALIPIX[3D] detector, with a Cu source and generator power at 45 kV and 40 mA.

Thermogravimetric analysis was performed on a TA Instruments TGA 5500. Samples were placed in a platinum pan and heated to 800 °C at a ramp rate of 5 °C min$^{-1}$ in a nitrogen atmosphere.

## Data availability

Data are available at Dryad (https://doi.org/10.5061/dryad.59zw3r2fb).

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

**Acknowledgements** This material is based on work supported by the National Science Foundation Graduate Research Fellowship Program under grant no. DGE-1656518. Any opinions, findings and conclusions or recommendations expressed in this material are those of the author(s) and do not necessarily reflect the views of the National Science Foundation. This work was supported, in whole or in part, by the Bill & Melinda Gates Foundation (no. INV-046940). Under the grant conditions of the Foundation, a Creative Commons Attribution 4.0 Generic License has already been assigned to the Author Accepted Manuscript version that might arise from this submission. Part of this work was performed at the Stanford Nano Shared Facilities, supported by the National Science Foundation under award no. ECCS-2026822. We thank D. Uruchurtu Patino and the Bao Group at Stanford for assistance with the tube furnace, as well as D. Ilyin and K. Hsiao for their technical support.

**Author contributions** J.M.K. and J.M.D. conceptualized the technique and experimental design. J.M.K. and L.R. developed the high-resolution system, furthered the custom printing programme and produced the particles herein shown. M.A.S. aided in materials characterization. J.M.K. wrote the paper with input from L.R., M.A.S., M.T.D. and J.M.D. All authors commented on and edited the paper.

**Competing interests** J.M.D. was a cofounder of, and has a financial stake in, Carbon, a CLIP-based 3D printing company. M.A.S. advises 3D Architech, a vat photopolymerization-based metal 3D printing company. L.R. cofounded EDO Additive Solutions, a 3D printing consulting firm. All other authors declare no competing interests.

**Additional information**
**Correspondence and requests for materials** should be addressed to Joseph M. DeSimone.

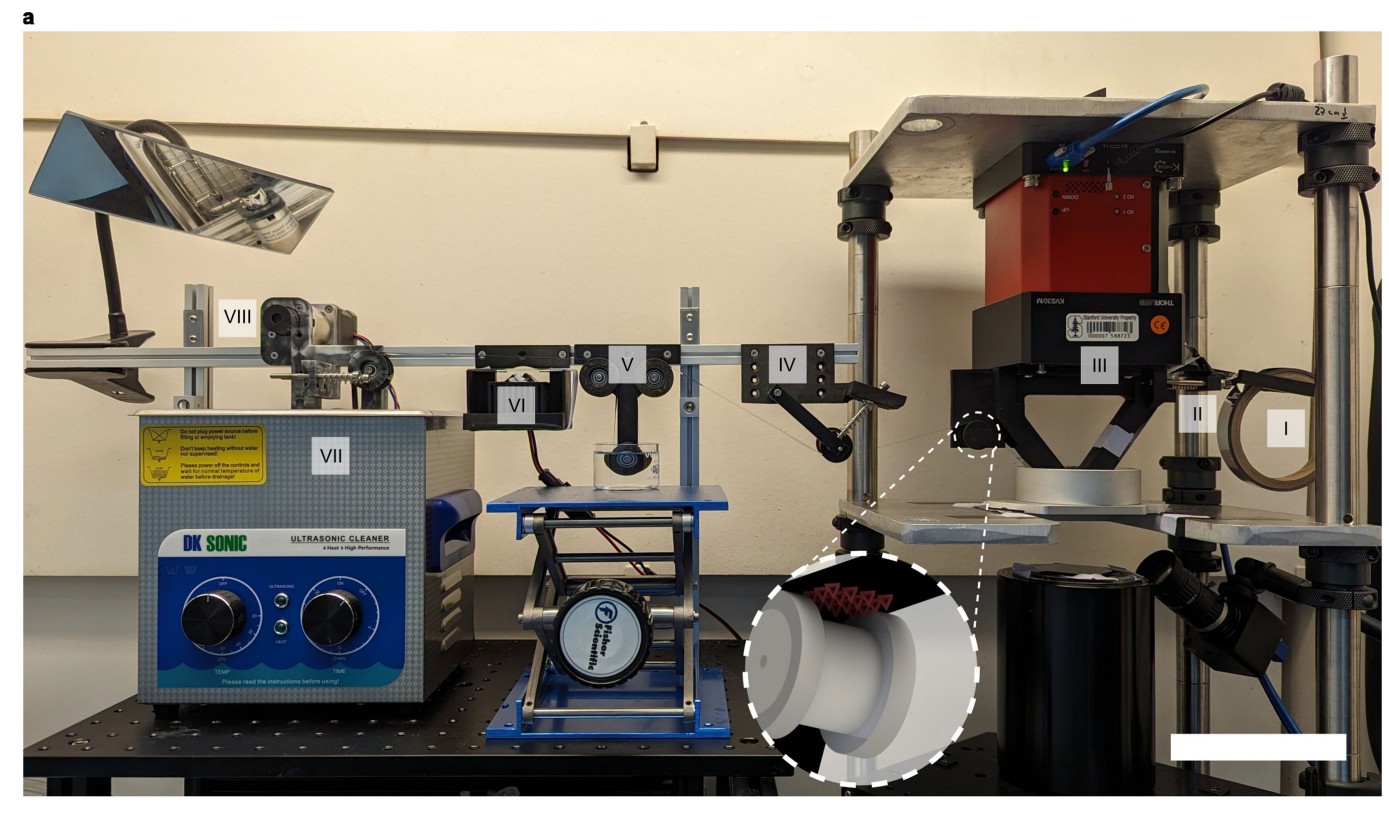

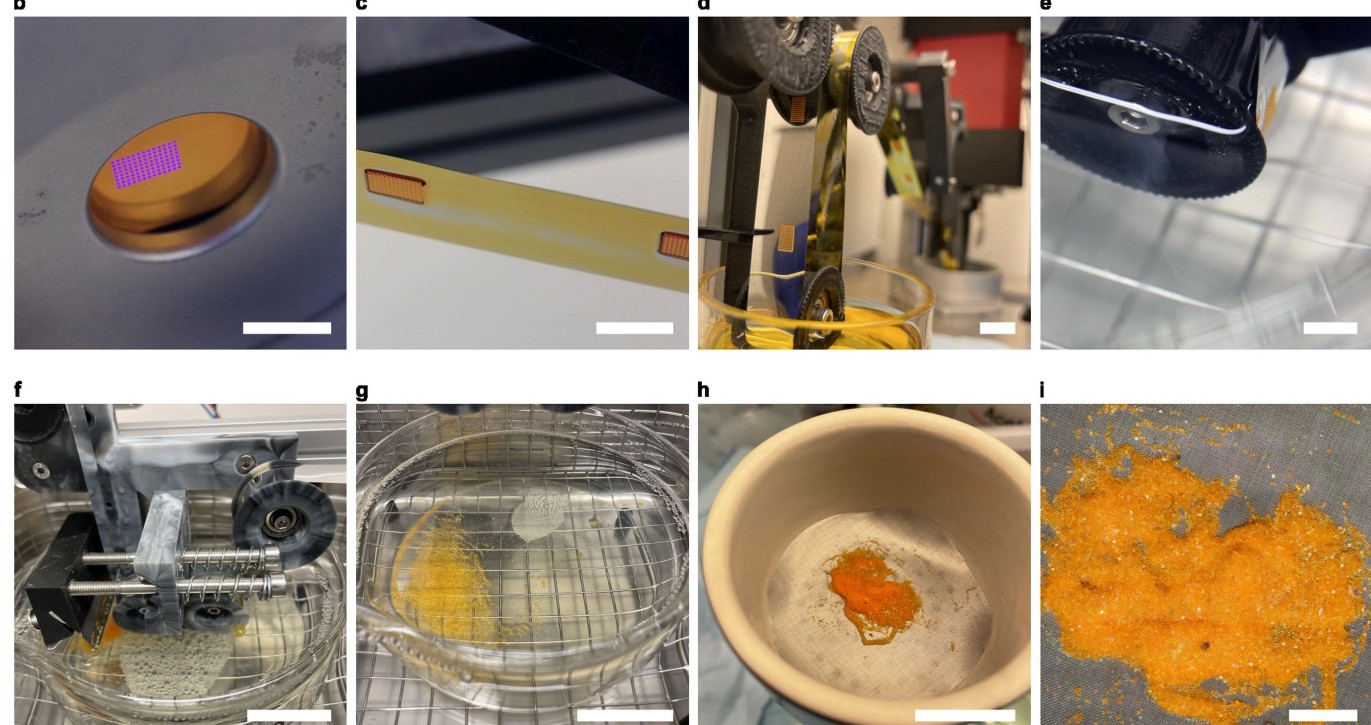

**Extended Data Fig. 1 | Experimental r2rCLIP mechanism and setup.**
As detailed in Fig. 1, the experimental r2rCLIP setup is comprised of a film unrolled from (I) a feed roll and (II) mechanically tensioned before passing over (III) a high precision *z* stage and CLIP assembly where the particle geometry is fabricated onto the film. The particles on film are then (IV) passed under a spring-tensioning system, (V) through isopropanol, (VI) through a secondary curing station (as necessary), and (VII) immersed in a nonionic surfactant solution in a heated sonication bath with or without a razor blade to induce

delamination. (VIII) The film is finally collected on a second roller. Images of particle arrays at several stages: **b**, during CLIP fabrication from the bottom of the vat looking through the Teflon AF window; **c**, immediately after exiting the vat; **d**, going into and immediately after cleaning; **e**, passing into the harvesting bath; **f**, in the harvesting mechanism (razor blade in retracted position); **g**, in the harvesting bath; **h**, during filtration over a mesh screen; and **i**, after initial filtration. Scale bars: **a**, 10 cm; **b**–**e**,**i**, 5 mm; **f**–**h**, 25 mm.

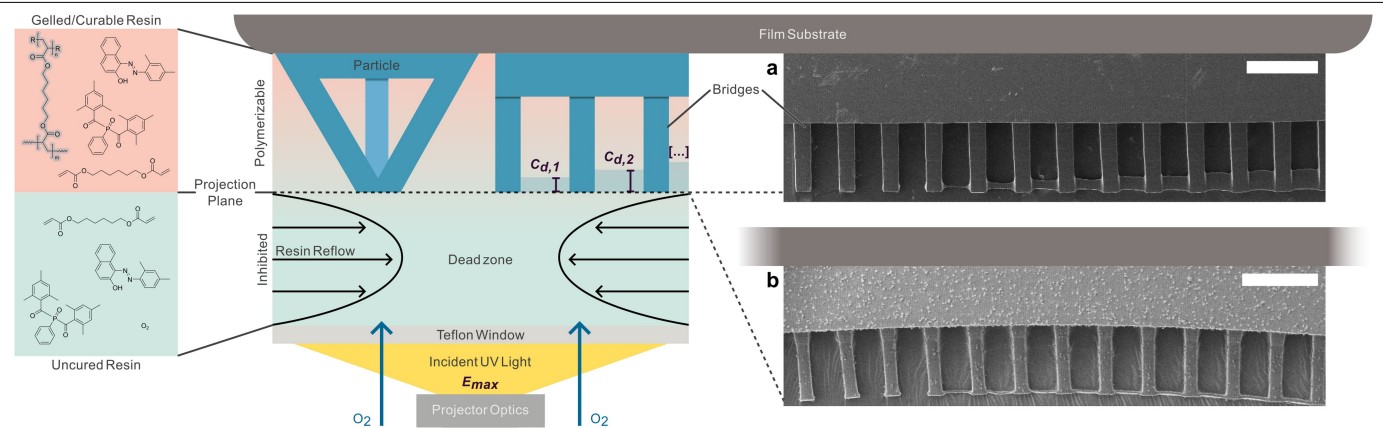

**Extended Data Fig. 2 | CLIP introduces a dead zone, inhibiting oxygen polymerization at the Teflon-resin interface.** Incident light passes through an optically clear Teflon AF window through which oxygen may simultaneous diffuse. Oxygen dominates immediately at this interface, yielding at a dead zone distance away to radical polymerization kinetics. Resin cured above this depth may be cured as here exemplified for the HDDA-HDDMA resin system components, a hollow tetrahedron particle, and an edge section of the working curve bridging test. Example substrate-oriented working curve bridge samples are shown for **a**, TMPTA and **b**, HDDA-Ceramic Mix resins for increasing delivered bridge dosage from left to right of 0.77 to 9.18 mJ/cm$^2$ and 16.07 to 41.32 mJ/cm$^2$ for the 12 bridge positions shown, respectively. Scale bars: 250 μm.

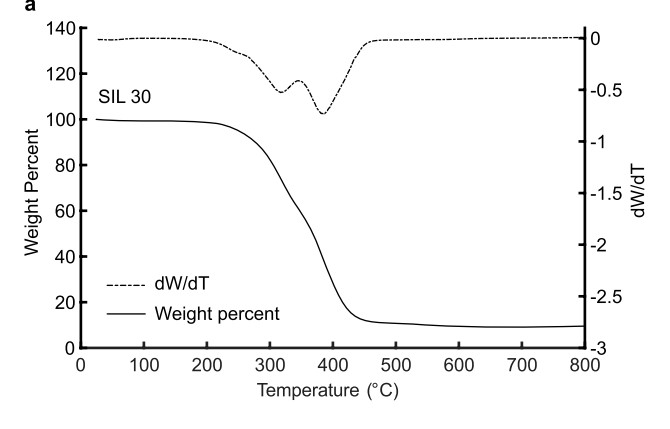

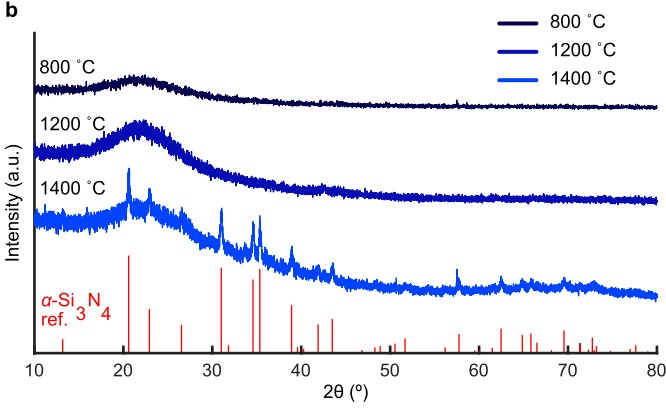

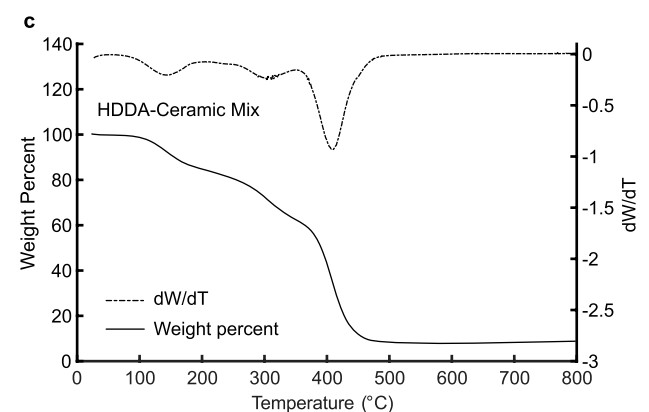

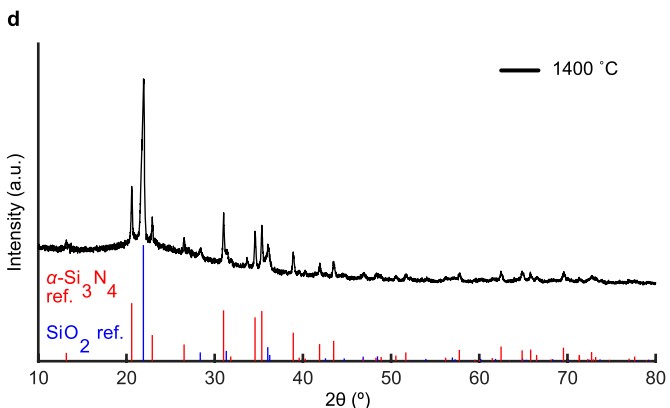

**Extended Data Fig. 3 | Characterization of preceramic resins.**
**a**, Thermogravimetric analysis (TGA) shows that after heating SIL 30 resin to 800 °C in nitrogen, 9.5% of the initial mass remains. **b**, X-ray diffraction patterns show that after pyrolysis of cured SIL30 resin at 800 °C and 1200 °C, the products remain amorphous. After annealing at 1400 °C, the measured pattern has reflections that match those for α-Si$_3$N$_4$ (PDF 04-0070851) from the

International Centre for Diffraction Data (ICDD)[51,52]. **c**, TGA of HDDA-Ceramic Mix shows that after heating to 800 °C in nitrogen, 8.8% of the initial mass remains. **d**, X-ray diffraction pattern showing that after pyrolysis of the HDDA-Ceramic mix at 1400 °C for a total of 8 h, the measured patterns has reflections that match those for α-Si$_3$N$_4$ (PDF 04-0070851) and SiO$_2$ (PDF 04-0075018) from the ICDD[53].