## [Peer Review File · Nature]

Manuscript Title: Roll-to-Roll, High-Resolution 3D Printing of Shape-Specific Particles

Reviewer Comments & Author Rebuttals

Reviewer Reports on the Initial Version:

Referee expertise:

Referee #1: structure-property-morphology-processing relationships of polymers

Referee #2: development of new polymer-based functional materials, 4D printing

Referees' comments:

Referee #1 (Remarks to the Author):

Summary: This manuscript describes an exciting avenue to print polymeric particles with controlled/precise geometries and micron-scale resolution, suggesting immense impact on drug and vaccine delivery, electronics, energy, and abrasive industries. This manuscript builds on earlier research related to CLIP printing processes with an integration of a non-static build platform that enables production of particles at high rates and scalability. The concept is demonstrated with commercially available compositions of diacrylates in combination with photo-radical initiators.

Originality and Significance: This innovation is novel for the literature and represents the continuation of the investigators' earlier discoveries, leveraging a unique combination of PRINT technologies together with their most recent manuscripts related to CLIP. This team is uniquely situated for this discovery.

Data and Methodology: The investigators combine imaging and a diversity of reagents to show versatility of the printing operation. The experiments clearly demonstrate the efficacy of the design of a novel printing operation. The experiments are well constructed and sufficient to demonstrate the novelty and versatility of the technique.

Statistical analysis: There was no concern for the lack of statistical treatment with adequate replicates and parameter space.

Conclusions: The investigators clearly offer robust conclusions, and appropriate experiments were performed to demonstrate the validity and reliability of this novel printing modality.

Suggested Improvements:

Further discussion of the importance of scale and relevance of the 1,000,000 particles is consistent with the scale to enable future technology. In other words, does the technologies of the future require this scale, does the scale match the technology needs?

The manuscript reads a bit too much as a review at the onset, and the authors should consider reducing the discussion of earlier work and devote more attention to the discovery. For example, more discussion of particle properties and process opportunities to tune polymer properties. Compositions that are relevant to the proposed broader impact would increase the impact of the manuscript. For example, can you print a drug delivery vehicle and demonstrate lack of toxicity or release rates as a function of polymer composition and shape/size.

References: Appropriate and comprehensive, leads reader to a clear progression of the literature.

Clarity and Context: Abstract and summary clearly points to the novel printing operation and its unique elements, and furthermore the summary focuses on printing scale, previously inaccessible geometries, and unique precursors for ceramics and other novel compositions. The conclusions are printing scale and particle geometry with a novel printing modality.

Referee #2 (Remarks to the Author):

DeSimone and coworkers present a new fabrication technique named as roll-to-roll continuous liquid interface production (r2rCLIP) for the fabrication of particles with controlled shapes and high resolution. This approach is based on this previous set up (high resolution CLIP, ref. Sci. Adv 2022) and replaced the static build with a continuous roll to roll system enabling the fast and fully automated particle fabrication.

Importantly, they demonstrate the versatility of the approach with a wide range of printable materials (commercial and non-commercial) from ceramics to soft hydrogels and can fabricate the impressive numbers of 1,000,000 particles per day. Also, they show the possibility of print complex geometries (including hollow ones) that cannot be manufactured by other means such as molding that can be varied in a single print.

The manuscript is clearly written and the references appropriate. Also, the quality of the display items is good.

Below a list of questions and suggestion for the authors:

- 1) The authors have tested the printability of 7 different commercial and non-commercial materials to prove the versatility of the technique. In terms of mechanical properties, the ceramic mix or the soft hydrogels are very different. To detach the printed particles from the PET film, they used tension and afterwards, ultrasonication. Do they observe damage with the softer materials? Does this roll-to-roll require adaptation of the parameters depending on the material employed? The

authors should comment on that.

2) The material used for the printed particles shown in Figures 1 and 3 is not clear. I assume this was done with the HDDA resin. Adding this data in the caption would help the reader.

3) The authors use the "bridging method" for measuring the resin curing properties at different exposures. In Figure 2, SEM images of the printed bridges are shown. However, one cannot see the complete bridge. Providing the image of the complete bridge and also the comparison with another material with less resolution (e.g. PEGDMA) would be interesting for the readers.

4) The authors show a great variety of printed geometries including a "drug delivery cube". For this, the box and the lid are printed separately and I am wondering how to control/manipulate these tiny containers to be open and closed.

5) Further investigation of the delivery depending on the geometry of the printed particles would be an asset and improve significantly the value of the publication.

Reviewer Comments:

Reviewer #1:

Summary: This manuscript describes an exciting avenue to print polymeric particles with controlled/precise geometries and micron-scale resolution, suggesting immense impact on drug and vaccine delivery, electronics, energy, and abrasive industries. This manuscript builds on earlier research related to CLIP printing processes with an integration of a non-static build platform that enables production of particles at high rates and scalability. The concept is demonstrated with commercially available compositions of diacrylates in combination with photo-radical initiators.

Originality and Significance: This innovation is novel for the literature and represents the continuation of the investigators' earlier discoveries, leveraging a unique combination of PRINT technologies together with their most recent manuscripts related to CLIP. This team is uniquely situated for this discovery.

Data and Methodology: The investigators combine imaging and a diversity of reagents to show versatility of the printing operation. The experiments clearly demonstrate the efficacy of the design of a novel printing operation. The experiments are well constructed and sufficient to demonstrate the novelty and versatility of the technique.

Statistical analysis: There was no concern for the lack of statistical treatment with adequate replicates and parameter space.

Conclusions: The investigators clearly offer robust conclusions, and appropriate experiments were performed to demonstrate the validity and reliability of this novel printing modality.

Suggested Improvements:

Further discussion of the importance of scale and relevance of the 1,000,000 particles is consistent with the scale to enable future technology. In other words, does the technologies of the future require this scale, does the scale match the technology needs?

We appreciate the reviewer's suggestion. We have broadened our discussion to elaborate on how the mass production of particles might align with the scale of future technological requirements. This expanded discussion aims to provide a more comprehensive understanding of the potential scalability of our technology.

The manuscript reads a bit too much as a review at the onset, and the authors should consider reducing the discussion of earlier work and devote more attention to the discovery. For example, more discussion of particle properties and process opportunities to tune polymer properties.

We appreciate the reviewer's feedback. We have revised our introductory discussion of previous work to shift our focus more towards our discovery.

We agree that a more detailed discussion of particle properties and opportunities to tune polymer properties would be beneficial. We have expanded this section (lines 171 – 209) to provide a more comprehensive understanding of our work.

Compositions that are relevant to the proposed broader impact would increase the impact of the manuscript. For example, can you print a drug delivery vehicle and demonstrate lack of toxicity or release rates as a function of polymer composition and shape/size.

We appreciate the reviewer's suggestion and have incorporated a discussion on compositions that hold relevance to the broader biological impact of particles fabricated via r2rCLIP (lines 184 – 195). A multitude of commercial photopolymer resin systems have been evaluated for biocompatibility and intended applications¹, with numerous research groups focusing on developing biocompatible resin systems for bio-scaffolds and similar delivery manifolds^{2,3}.

Furthermore, the aspects of materials biocompatibility, cytotoxicity, influence of shape and size on localization and delivery, and translational dosing applications have been previously explored in the context of drug delivery vehicles⁴⁻¹³. This body of work in the micro-vehicle drug-delivery field lays a robust foundation for further studies involving r2rCLIP-produced delivery vehicles.

We share the excitement about this potential and direction, and plan to delve into this area in our future work. However, we believe that kinetic release studies extend beyond the scope of this work, which is to introduce our new particle production / synthesis methodology.

References: Appropriate and comprehensive, leads reader to a clear progression of the literature.

Clarity and Context: Abstract and summary clearly points to the novel printing operation and its unique elements, and furthermore the summary focuses on printing scale, previously inaccessible geometries, and unique precursors for ceramics and other novel compositions. The conclusions are printing scale and particle geometry with a novel printing modality.

Reviewer #2:

DeSimone and coworkers present a new fabrication technique named as roll-to-roll continuous liquid interface production (r2rCLIP) for the fabrication of particles with controlled shapes and high resolution.

This approach is based on this previous set up (high resolution CLIP, ref. Sci. Adv 2022) and replaced the static build with a continuous roll to roll system enabling the fast and fully automated particle fabrication.

Importantly, they demonstrate the versatility of the approach with a wide range of printable materials (commercial and non-commercial) from ceramics to soft hydrogels and can fabricate the impressive numbers of 1,000,000 particles per day. Also, they show the possibility of printing complex geometries (including hollow ones) that cannot be manufactured by other means such as molding that can be varied in a single print.

The manuscript is clearly written and the references appropriate. Also, the quality of the display items is good.

Below is a list of questions and suggestions for the authors:

1) The authors have tested the printability of 7 different commercial and non-commercial materials to prove the versatility of the technique. In terms of mechanical properties, the ceramic mix or the soft hydrogels are very different. To detach the printed particles from the PET film, they used tension and afterwards, ultrasonication. Do they observe damage with the softer materials? Does this roll-to-roll require adaptation of the parameters depending on the material employed? The authors should comment on that.

We appreciate the reviewer's interest in the process. In response to this, we have included a discussion on the changes in parameter considerations based on the materials used during the r2rCLIP process in the Supplementary Information section (see *Parameter Optimization and Delamination in Supplementary Information*). This section now provides a detailed account of our qualitative observations and the subsequent optimizations we introduced to the system during the development of the process. These modifications were instrumental in achieving the range of soft to hard materials demonstrated in our study.

2) The material used for the printed particles shown in Figures 1 and 3 is not clear. I assume this was done with the HDDA resin. Adding this data in the caption would help the reader.

We appreciate the reviewer's request for clarification. We have now added a note to both Figure 1 and Figure 3 to specify that the resin system used in these figures is HDDA-based, as correctly noted by the reviewer.

3) The authors use the "bridging method" for measuring the resin curing properties at different exposures. In Figure 2, SEM images of the printed bridges are shown. However, one cannot see the

complete bridge. Providing the image of the complete bridge and also the comparison with another material with less resolution (e.g. PEGDMA) would be interesting for the readers.

We appreciate the reviewer's suggestion to expand Figure 2. We have now included several bridge series from resins of increasing penetration depth in the figure. This addition visually demonstrates how the thickness of the bridges increases more rapidly at the same dosage.

Furthermore, we have updated Extended Data Figure 2 to include a Scanning Electron Microscopy (SEM) image of a portion of a bridge set. The orientation of this image is designed to provide a visual connection between the graphic and the experimental resulting bridges.

4) The authors show a great variety of printed geometries including a "drug delivery cube". For this, the box and the lid are printed separately and I am wondering how to control/manipulate these tiny containers to be open and closed.

We appreciate the reviewer's inquiry. Similar to the cited report from the Langer Lab⁴, the containers were manually filled and the caps were manually placed onto the containers to illustrate the process.

We have added a section to the methods (*Cargo Delivery Cube Fabrication in Methods*) and main body (lines 184 – 195) of the paper to explain our approach in this study and our plans for future work in fabricating drug delivery vehicles. We hope this provides a clearer understanding of our process and future direction.

5) Further investigation of the delivery depending on the geometry of the printed particles would be an asset and improve significantly the value of the publication.

We appreciate the reviewer's suggestion and share the enthusiasm for exploring the potential of r2rCLIP fabricated drug delivery vehicles in translational applications. However, we consider that a detailed investigation of this aspect would exceed the scope of this paper, which primarily focuses on introducing the r2rCLIP process and its mechanisms.

It's worth noting that the release mechanisms, parameters, biologically imperative shape-dependencies, and potential of similar drug delivery cubes and molded particles have been extensively demonstrated in previous studies⁴⁻¹³. These works lay a solid foundation for our technology to build upon and discussion of such has now been included in the manuscript (lines 184 – 195).

We are indeed excited about this direction and are concurrently investigating it for future research papers concerning r2rCLIP technology. We believe this will provide a more appropriate platform to delve into the translational applications of r2rCLIP fabricated drug delivery vehicles.

References

1. Guttridge, C., Shannon, A., O'Sullivan, A., O'Sullivan, K. J. & O'Sullivan, L. W. Biocompatible 3D printing resins for medical applications: A review of marketed intended use, biocompatibility certification, and post-processing guidance. *Annals of 3D Printed Medicine* **5**, 100044 (2022).
2. Xu, X. *et al.* Vat photopolymerization 3D printing for advanced drug delivery and medical device applications. *Journal of Controlled Release* **329**, 743–757 (2021).
3. Wang, J. *et al.* Emerging 3D printing technologies for drug delivery devices: Current status and future perspective. *Advanced Drug Delivery Reviews* **174**, 294–316 (2021).
4. McHugh, K. J. *et al.* Fabrication of fillable microparticles and other complex 3D microstructures. *Science* **357**, 1138–1142 (2017).
5. Sarmadi, M. *et al.* Experimental and computational understanding of pulsatile release mechanism from biodegradable core-shell microparticles. *Science Advances* **8**, eabn5315 (2022).
6. Sadeghi, I., Lu, X., Sarmadi, M., Langer, R. & Jaklenec, A. Micromolding of Thermoplastic Polymers for Direct Fabrication of Discrete, Multilayered Microparticles. *Small Methods* **6**, 2200232 (2022).
7. Lu, X. *et al.* Engineered PLGA microparticles for long-term, pulsatile release of STING agonist for cancer immunotherapy. *Science Translational Medicine* **12**, eaaz6606 (2020).
8. Shukla, S. K., Sarode, A., Wang, X., Mitragotri, S. & Gupta, V. Particle shape engineering for improving safety and efficacy of doxorubicin — A case study of rod-shaped carriers in resistant small cell lung cancer. *Biomaterials Advances* **137**, 212850 (2022).
9. Rolland, J. P. *et al.* Direct Fabrication and Harvesting of Monodisperse, Shape-Specific Nanobiomaterials. *J. Am. Chem. Soc.* **127**, 10096–10100 (2005).
10. Mathaes, R., Winter, G., Besheer, A. & Engert, J. Non-spherical micro- and nanoparticles: fabrication, characterization and drug delivery applications. *Expert Opinion on Drug Delivery* **12**, 481–492 (2015).

11. Gratton, S. E. A. *et al.* Nanofabricated particles for engineered drug therapies: A preliminary biodistribution study of PRINT™ nanoparticles. *Journal of Controlled Release* **121**, 10–18 (2007).
12. Fernandes, R. & Gracias, D. H. Self-folding polymeric containers for encapsulation and delivery of drugs. *Advanced Drug Delivery Reviews* **64**, 1579–1589 (2012).
13. Euliss, L. E., DuPont, J. A., Gratton, S. & DeSimone, J. Imparting size, shape, and composition control of materials for nanomedicine. *Chem. Soc. Rev.* **35**, 1095–1104 (2006).

Reviewer Reports on the First Revision:

Referees' comments:

Referee #1 (Remarks to the Author):

The revisions were thoughtful and comprehensive.

Referee #2 (Remarks to the Author):

The authors have have adequately addressed all the comments made by the reviewers in the revised version of the manuscript. The additional information added regarding the process' details and the materials is very valuable for the readers and potential future users of this approach.

Therefore, recommend for publication without further changes.